# Synchronous and asynchronous modes of synaptic transmission utilize different calcium sources

Hua Wen[1][†], Jeffrey M Hubbard[1][†][‡], Benjamin Rakela[1], Michael W Linhoff[1], Gail Mandel[1,2], Paul Brehm[1]*

[1]Vollum Institute, Oregon Health and Science University, Portland, United States; [2]Howard Hughes Medical Institute, Oregon Health and Science University, Portland, United States

**Abstract** Asynchronous transmission plays a prominent role at certain synapses but lacks the mechanistic insights of its synchronous counterpart. The current view posits that triggering of asynchronous release during repetitive stimulation involves expansion of the same calcium domains underlying synchronous transmission. In this study, live imaging and paired patch clamp recording at the zebrafish neuromuscular synapse reveal contributions by spatially distinct calcium sources. Synchronous release is tied to calcium entry into synaptic boutons via P/Q type calcium channels, whereas asynchronous release is boosted by a propagating intracellular calcium source initiated at off-synaptic locations in the axon and axonal branch points. This secondary calcium source fully accounts for the persistence following termination of the stimulus and sensitivity to slow calcium buffers reported for asynchronous release. The neuromuscular junction and CNS neurons share these features, raising the possibility that secondary calcium sources are common among synapses with prominent asynchronous release.

*For correspondence: brehmp@ohsu.edu

[†]These authors contributed equally to this work

[‡]Present address: Brain and Spine Institute (ICM), Paris, France

Competing interests: The authors declare that no competing interests exist.

## Introduction

Physiological studies have pointed increasingly to a central role played by asynchronous release in mediating synaptic transmission (*Goda and Stevens, 1994*; *Lu and Trussell, 2000*; *Hefft and Jonas, 2005*; *Iremonger and Bains, 2007*; *Best and Regehr, 2009*). At most synapses, the asynchronous contribution to release is smaller than the synchronous component, but becomes more prominent with repetitive stimulation (*Atluri and Regehr, 1998*; *Lu and Trussell, 2000*; *Hefft and Jonas, 2005*). Little is known about the functional significance, but the most widely held idea is that it provides persistent transmitter release under conditions where phase locking to the action potential is not required (*Atluri and Regehr, 1998*; *Hefft and Jonas, 2005*; *Best and Regehr, 2009*). At the zebrafish neuromuscular junction (NMJ), asynchronous release may augment release during bouts of prolonged swimming, where release probability may be severely compromised.

Synchronous and asynchronous modes of release are thought to arise from different calcium dynamics surrounding the presynaptic release zones. According to this idea, synchronous release is triggered by highly localized calcium transients resulting from the opening of calcium channels near the active zones (*Adler et al., 1991*; *Stanley, 1993*; *Neher, 1998*; *Meinrenken et al., 2002*; *Augustine et al., 2003*; *Eggermann et al., 2012*), whereas a more slowly decaying component of intracellular calcium is proposed to account for the asynchronous release (*Rahamimoff and Yaari, 1973*; *Goda and Stevens, 1994*; *Cummings et al., 1996*; *Atluri and Regehr, 1998*; *Chen and Regehr, 1999*). The slow accumulation of calcium during repeated stimulation can potentially account for both the delayed onset and persistence (*Lu and Trussell, 2000*; *Hefft and Jonas, 2005*). Much of the evidence for this

**eLife digest** Neurons communicate with one another at junctions called synapses. The arrival of an electrical signal known as an action potential at the first (presynaptic) neuron causes calcium ions to flood into the cell. This in turn causes the neuron to release packages of chemicals called neurotransmitters into the synapse. These activate receptors on the second (postsynaptic) neuron, triggering a new action potential that travels down the axon to the next synapse.

The ions that trigger the release of the neurotransmitters are thought to enter the neuron through special calcium channels on or near the synapse. A sudden discrete influx of calcium ions causes the neuron to release many packages of transmitter simultaneously. This is called synchronous release. By contrast, when successive action potentials occur in the same neuron, the ions entering through the calcium channels accumulate inside the cell. This is thought to account for a sustained release of the neurotransmitter that continues even in the absence of nerve action potentials. This is called asynchronous release.

Wen et al. have now obtained evidence that these two forms of release might be triggered by calcium from different sources. The work was performed using a synapse between nerve and muscle cells in zebrafish: it has been shown that channels called P/Q calcium channels control the release of neurotransmitters at this synapse in zebrafish.

Mutant zebrafish with greatly reduced numbers of P/Q channels showed reduced synchronous release, but normal asynchronous release. Blocking the P/Q channels with a specific toxin in normal zebrafish eliminated synchronous release but left asynchronous release intact. Imaging experiments on these toxin-treated zebrafish revealed that a wave of calcium ions that propagated from a distant source coincided with the onset of asynchronous release. This wave of calcium fully accounted for the delayed onset and the persistence of asynchronous release following termination of the action potentials. This study further demonstrates that asynchronous release can be triggered by calcium ions that do not enter through the P/Q calcium channels.

Waves of calcium have been described in the nervous system before, but their significance has always been unclear. The work of Wen et al. offers the first possible explanation for the role of these waves, and further experiments are now needed to determine whether this process happens at other types of synapses.

idea rests on the observation that the slow calcium buffer EGTA can block asynchronous release while leaving synchronous release intact (*Adler et al., 1991*; *Cummings et al., 1996*; *Atluri and Regehr, 1998*; *Lu and Trussell, 2000*). The fact that the faster calcium buffer BAPTA is required to inhibit synchronous release has been interpreted to represent a more restricted calcium domain that is in close vicinity to the calcium sensors underlying exocytosis (*Neher, 1998*; *Eggermann et al., 2012*).

There has been little consideration of alternative secondary sources of calcium, despite evidence for their involvement in synaptic transmission (*Collin et al., 2005*; *Berridge, 2006*). In one recent case however, an unexpected source of calcium for asynchronous release was reported in the form of an unusual voltage dependent calcium channel type that provides persistent calcium entry (*Few et al., 2012*). Our investigation into possible separate sources of calcium in zebrafish motor neurons was prompted by our observation that a block of calcium entry through the presynaptic P/Q calcium channels fully inhibited synchronous release, leaving asynchronous release intact. Zebrafish NMJ offers a unique opportunity to explore the temporal and spatial relationships between calcium entry and the two release modes through combining paired recording and live calcium imaging. We now present evidence for a source of off-synapse calcium that triggers the initiation of asynchronous release. This novel calcium source for asynchronous release predicts many of the central features for this mode including sensitivity to calcium buffers, persistence following termination of the stimulus, and non-phase locking to the presynaptic action potential.

## Results

Paired patch clamp recordings from the caudal primary motor neuron (CaP) and target fast skeletal muscle showed a stereotypic transition from exclusively synchronous to principally asynchronous transmission when stimulated at frequencies greater than 20 Hz (*Figure 1A,B*; *Wen et al., 2010*). Behaviorally

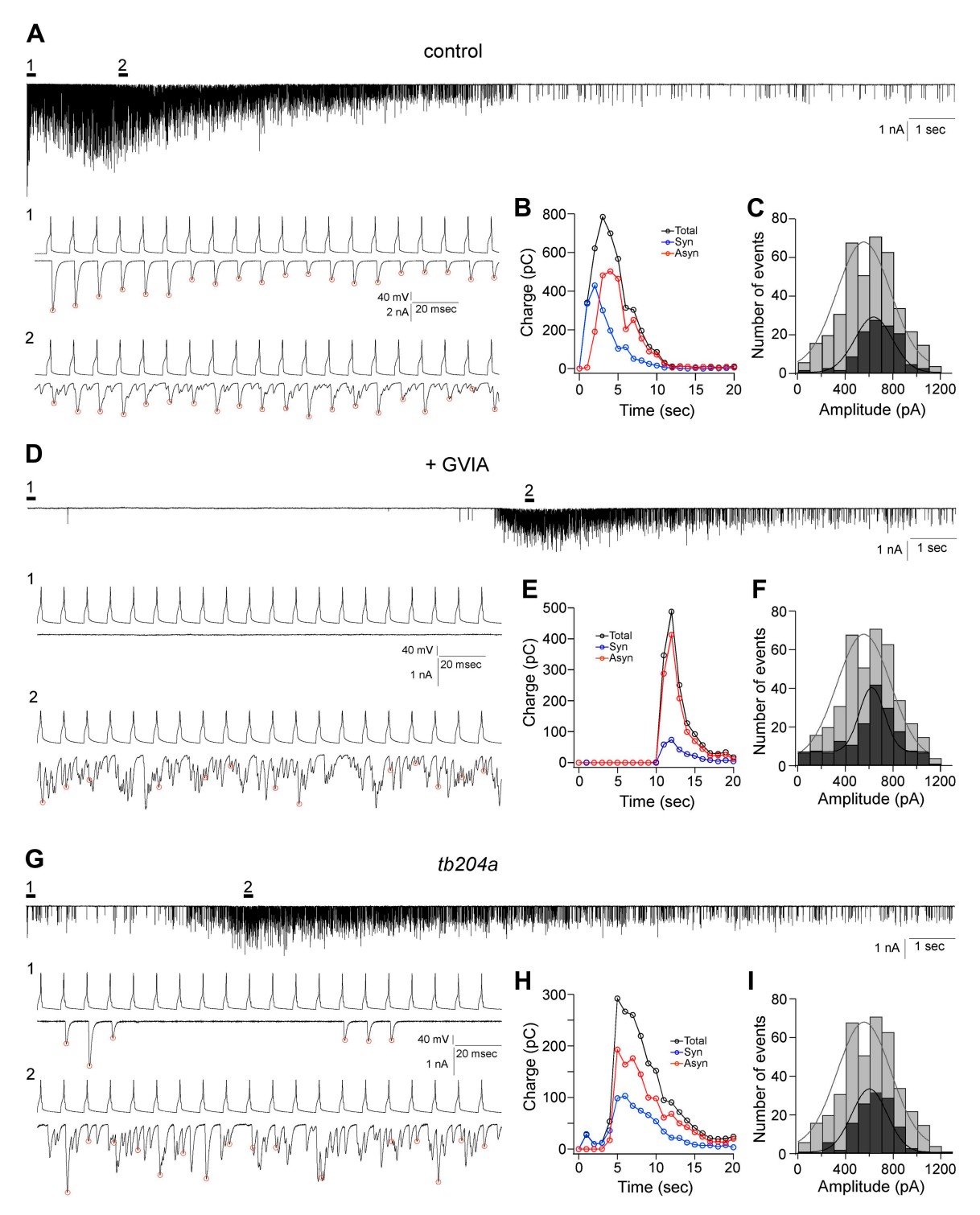

**Figure 1**. Asynchronous synaptic transmission remains intact in the P/Q calcium channel mutant *tb204a* and following treatment of wild-type fish with ω-conotoxin GVIA. (**A–C**) A representative paired recording from untreated wild-type fish. (**A**) Voltage clamp traces of EPCs in response to 20 s, 100 Hz stimulation of the motor neuron. Expanded views with both action potentials and associated postsynaptic EPCs showing early synchronous (**A1**) and mixed synchronous and asynchronous release at the peak of release (**A2**). (**B**) Quantitation of the time-dependence of synchronous (blue), asynchronous (red) and total (black) synaptic charge integrals determined using the methods described in *Wen et al. (2010)*. (**C**) Comparison of the stimulus evoked asynchronous event amplitudes recorded during the last 10 s of stimulation (black fill) and spontaneous synaptic current amplitudes (gray fill, 402 events

*Figure 1. Continued on next page*

*Figure 1. Continued*

from 17 cells). The distributions are fit by a Gaussian function with means corresponding to 637 pA and 556 pA. (**D–F**) A representative paired recording from fish treated with 1 µM ω-conotoxin GVIA. (**D**) Traces of EPCs with expanded views showing near elimination of synchronous release (**D1**) and intact asynchronous release (**D2**) in ω-conotoxin GVIA-treated fish. (**E**) Time course of release for the recording shown in **D**. (**F**) Comparison of its asynchronous event amplitude (black fill) and the same spontaneous synaptic current amplitudes used for 1C and 1I (gray fill). Events during the last 5 s of stimulation were included in the analysis. The mean value from a Gaussian fit for ω-conotoxin GVIA-treated fish was 620 pA. (**G–I**) A representative paired recording from the mutant line *tb204a*. (**G**) Traces of action potentials and EPCs from a homozygous *tb204a* mutant showing greatly reduced synchronous release (**G1**) and intact late asynchronous release (**G2**). (**H**) The time course of release for the recording shown in **G**. (**I**) Comparison of its asynchronous event amplitudes (black fill) and the spontaneous synaptic current amplitudes (gray fill). Events during the last 5 s of stimulation were included in the analysis. The mean value from a Gaussian fit for the mutant was 601 pA. Red circles in (**A**), (**D**), and (**G**) mark the peaks of synchronous events. All experiments were performed with 5 mM EGTA in the intracellular solution.

evoked contractures of zebrafish axial muscle correspond to frequencies between 20 Hz and 100 Hz, so we continue to use the latter stimulus frequency as the benchmark for our studies. At 100 Hz, greater than 95% of the synaptic responses were phase locked to the presynaptic action potential during the first second of stimulation (*Figure 1A1,B*; *Wen et al., 2010*). The onset of asynchronous release occurred after the first second of stimulation, displaying a time-dependent increase in overall contribution during the ensuing stimulation (*Figure 1A2,B*). The release was quantitated as charge transfer by integrating the EPCs for each consecutive second of the stimulation, and synchronous and asynchronous events were separated on the basis of their timing to the action potential (*Wen et al., 2010*). The release associated with synchronous vs asynchronous events showed a time-dependent transition, and overall each of the two modes accounted for approximately half of the total synaptic transmission (*Figure 1B*; *Wen et al., 2010*). The amplitudes of the asynchronous events were indistinguishable from the spontaneous synaptic events measured in the absence of stimulation (*Figure 1C*), consistent with each representing individual quanta. The synchronous release is strictly dependent on P/Q-type calcium channel function (*Figure 1D*; *Wen et al., 2013*). Inhibiting P/Q calcium channels with 1 µM ω-conotoxin GVIA nearly abolished synchronous release (*Figure 1D1,E*). Unexpectedly, asynchronous release remained intact in the ω-conotoxin GVIA treated fish, but with a greatly delayed onset compared to the control fish (*Figure 1D2,E*). Quantitatively, over 85% of release seen in ω-conotoxin GVIA fish was associated with the asynchronous mode, and this likely represents an underestimate because of the resolution of our analysis (*Figure 1E*). Similar to the control, the amplitude of the asynchronous events was indistinguishable from spontaneous miniature events (*Figure 1F*). The motility mutant line *tb204a* has greatly compromised P/Q calcium channel function but is not a complete null (*Wen et al., 2013*). Accordingly, the synchronous release was reduced but not eliminated completely (*Figure 1G1*), leaving asynchronous release intact (*Figure 1G2*). Quantitation of the time-dependent contributions showed both reduced synchronous release and delayed onset of asynchronous release for *tb204a* compared to control (*Figure 1H*). Once again, the amplitude of the late asynchronous event class was indistinguishable from the spontaneous events measured in the absence of stimulation (*Figure 1I*), as well as those asynchronous events recorded from control (*Figure 1C*) and ω-conotoxin GVIA-treated (*Figure 1F*) fish.

When expressed as the time required to reach peak response during the stimulus train, the values were largest for ω-conotoxin GVIA-treated, smallest for control, and intermediate for *tb204a* mutant fish (*Figure 2A,B*). The time to peak release for ω-conotoxin GVIA-treated fish and *tb204a* mutant were much more variable than seen in control fish, but despite the variability both were significantly prolonged when compared to control ($p < 0.001$; *Figure 2B*).

Evidence for delayed asynchronous release in the presence of ω-conotoxin GVIA was next examined using an optical indicator of exocytosis. For this purpose, we recorded stimulus-driven fluorescence changes in a transgenic line of fish expressing synaptopHluorin in the spinal motor neurons. Exocytosis was signaled by an increase in fluorescence that results from exposure of the vesicular synaptopHluorin protein to neutral pH (*Miesenbock et al., 1998*). Fluorescence was monitored in a single image plane that was selected prior to stimulation on the basis of distribution and number of boutons. For this purpose each CaP neuron was filled with the fluorescent Alexa Fluor 647 dye via the patch pipette (*Figure 3A*, fill). Stimulation of CaP motor neurons for 10 s at 100 Hz resulted in a robust fluorescence increase that was largely restricted to the boutons associated with synapses (*Figure 3A*, spH). The synaptic location of the regions of interest (ROIs) was determined by means of the postsynaptic receptor

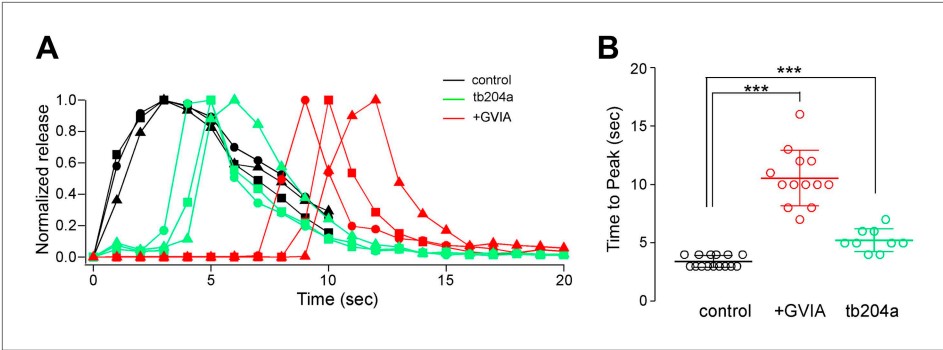

**Figure 2**. Overall comparisons of release time course between control, ω-conotoxin GVIA-treated and *tb204a* mutant fish. (**A**) The time courses for paired recordings from wild-type (black), *tb204a* mutant (green) and ω-conotoxin GVIA-treated (red) fish. The total release was expressed as the integrated charge for each consecutive second of the recording and normalized for comparison. (**B**) Scatter plot for recordings from wild type (black; 3.4 ± 0.5 s, n = 15), *tb204a* mutant (green; 5.2 ± 1.0 s, n = 9) and ω-conotoxin GVIA-treated (red; 10.5 ± 2.4 s, n = 13) fish comparing the time to peak release. Bars indicate the mean value and SD for each group. Asterisks indicate p<0.001. All experiments were performed with 5 mM EGTA in the intracellular solution.

label α−bungarotoxin (*Figure 3A*, α-btx and overlay). Next, stimulus-driven changes in synaptopHluorin fluorescence were determined. Images were captured at 33 ms intervals in a single plane that contained candidate ROIs. Post stimulus, images were corrected by background subtraction and then each ROI was quantitated as the ratio of delta fluorescence to prestimulus fluorescence ($\Delta F/F_0$, *Figure 3B*). This corrects for the expression of the pHluorin on the surface and in subcellular compartments other than vesicles. The fluorescence for control ROIs rose immediately at the start of 100 Hz stimulation and reached a plateau during the subsequent few seconds of stimulation (gray traces, *Figure 3B*). However, the fluorescence signal in ω-conotoxin GVIA-treated neurons showed greatly delayed onset of fluorescence for comparable ROIs (color traces, *Figure 3B*). Additionally, the synaptopHluorin signal was reduced in the ω-conotoxin GVIA-treated fish (*Figure 3B*). Comparisons of the onset of fluorescence increase for overall ROIs were made by measuring the time required to reach 50% maximal intensity from the initiation of stimulation. The distribution showed non-overlapping values for control and ω-conotoxin GVIA-treated fish (*Figure 3C*). These data agree well with those obtained by paired recordings showing delayed onset of asynchronous synaptic transmission in the absence of P/Q type calcium channel function (*Figures 1 and 2*).

The delayed onset of asynchronous release seen with paired recording and synaptopHluorin imaging prompted the exploration into spatial changes in stimulus-driven calcium levels. For this purpose, calcium indicators Fluo-4 or Fluo-5F were loaded (*Figure 4*, Fluo-4) along with Alexa Fluor 647 for post experimental three-dimensional morphological reconstruction of the CaP neuron (*Figure 4*, fill). As with synaptopHluorin measurements, calcium imaging required acquisition speed that restricted measurements to a single image plane. With a 40× objective the plane of focus usually included the soma, axon initial segment, a large region of axon with the major branch point, and a field of synaptic boutons. Presynaptic ROIs were established on the basis of postsynaptic α-btx labeling (*Figure 4*, α-btx). When stimulated at 100 Hz, time-dependent increases in fluorescence included soma, axon and boutons (*Figure 4*, Fluo-4). This location of distal fluorescence co-localized principally with the α-btx (*Figure 4*, merge).

This approach was then used to determine time-dependent changes in calcium for both control (*Figure 5A,B*) and ω-conotoxin GVIA-treated (*Figure 5C,D*) fish in response to 100 Hz stimulation. Fluorescence change was determined for each ROI and expressed as the ratio of the green fluorescence jump ($\Delta G$, $Ca^{2+}$ signal) to red fluorescence (R, fill). This method allowed us to normalize the fluorescence change to differences in cell volume (*Figure 5B,D*). In control neurons, $\Delta G/R$ within the axon initial segment and distal boutons showed no significant difference in the timing of signal onset (*Figure 5A,B*). By contrast, in fish treated with ω-conotoxin GVIA, the fluorescent signal onset was delayed for all regions except the axon initial segment (*Figure 5C,D*). Moreover, the delay was longer with greater distance from the soma (*Figure 5C,D*). The time of onset for the calcium signal was considerably briefer

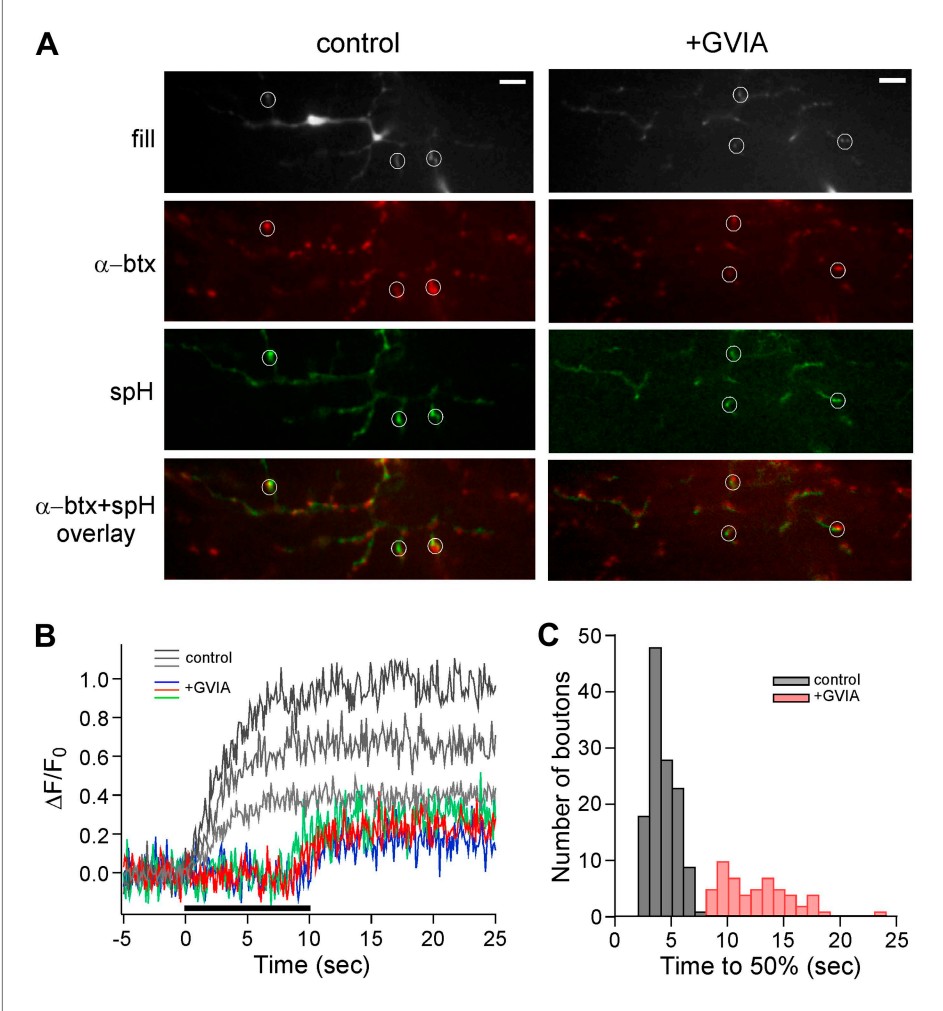

**Figure 3**. Delayed release in ω-conotoxin GVIA-treated fish is also observed by means of the exocytotic indicator synaptopHluorin. (**A**) Images taken from a single focal plane of the CaP motor neuron terminals showing the motor neuron fill with Alexa Fluor 647 (gray), postsynaptic labeling with α-btx (red), peak stimulus-induced synaptopHluorin fluorescence (green) and α-btx/synaptopHluorin overlay. These images are shown for a single control (left) and ω-conotoxin GVIA-treated (right) fish. The scale bar corresponds to 10 μm. (**B**) Stimulus-driven fluorescence change, expressed as $\Delta F/F_0$ for the three representative boutons shown for control (gray) and ω-conotoxin GVIA treated (colored) synaptopHluorin motor neurons. The synaptopHluorin signal was measured at each ROI during the time course of 100 Hz stimulation (indicated by the black bar in **B**). (**C**) The histogram showing time required to reach 50% maximal fluorescence increase for boutons of control (gray, 127 boutons from 8 fish) and ω-conotoxin GVIA-treated fish (red, 55 boutons from 4 fish). Experiments were performed with 5 mM EGTA in the intracellular solution.

than the asynchronous release shown in *Figure 1* because of the differences in EGTA concentrations used. The differences between 0.5 mM and 5 mM EGTA on the signal onset are later dealt with in the results. The entire videos from which the static images in *Figure 5* were extracted are available as *Video 1* and *Video 2*.

The delayed onset of calcium indicator signal in boutons of ω-conotoxin GVIA-treated fish pointed to a process involving delayed rise in intracellular calcium at the boutons following stimulation at the soma. However, it was necessary to exclude the possibility that the calcium indicator was competing with the endogenous calcium handling in the cell. To address this possibility, we compared two calcium indicators, 100 μM Fluo-4 ($k_d$ = 345 nM) to 100 μM Fluo-5F ($k_d$ = 2.3 μM), which have approximately sevenfold different binding constants for calcium (*Figure 6*). Comparisons were made on the basis of the onset of calcium signal in the boutons in both control and ω-conotoxin GVIA-treated fish. With

**Figure 4**. Stimulus-driven calcium signals in CaP motor neuron terminals occurred at synaptic boutons. The fill corresponds to a maximal intensity projection image of the motor neuron filled with Alexa Fluor 647. An arrowhead indicates the soma. A single plane of focus in the filled neuron showing postsynaptic α-btx label, peak Fluo-4 calcium signal and merged α-btx and Fluo-4 signal. The scale bar corresponds to 10 µm.

both Fluo-4 and Fluo-5F the time to 20% rise was significantly longer in ω-conotoxin GVIA treated fish compared to control fish (*Figure 6*). Importantly, there was no significant difference between the dyes in ω-conotoxin GVIA-treated fish (*Figure 6*). The time to 20% rise was marginally different (p=0.01) between the dyes in control experiments, potentially reflecting the increased time required for the lower affinity calcium indicators to measure the calcium rise.

To determine whether the delay in calcium signal onset correlated with the physical distance of the boutons from the proximal axon or soma, we combined calcium imaging with morphological reconstruction of the motor neuron. For this purpose each motor neuron was reconstructed as a three-dimensional image using Imaris filament software on the basis of the dye fill (*Figure 7C*). A 63× objective was substituted to obtain greater detail of the synaptic boutons resulting in exclusion of the soma from the field of view. Therefore, a reference point for distance zero was chosen that was approximately 80 µm from the soma where the axon has traversed the notochord (*Figure 7C*, arrow head). The time-dependent changes in fluorescence were first determined for a series of ROIs corresponding to hot spots at different distances from the reference point (*Figure 7A,B*). The rise in fluorescence was delayed with greater distance from the soma (*Figure 7B*). When expressed as time to 20% rise vs distance from the reference point, a striking quasi-linear relationship was observed for the boutons (*Figure 7D*). This relationship was determined for all of the recordings performed in ω-conotoxin GVIA-treated neurons (*Figure 7E*). The mean velocity of travel obtained in boutons determined for the cumulative data was 74 µm/s in ω-conotoxin GVIA-treated fish. By contrast, the relationships obtained for control neurons showed no dependence on distance in keeping with velocity of the propagating action potential (*Figure 7E*). In ω-conotoxin GVIA-treated neurons the distance dependence was not observed for the proximal axon, suggesting that the signaling underlying calcium propagation in these regions may be different from that of the distal terminals (*Figure 7D,E*).

An alternative possibility to calcium propagation as causal to the delayed release and apparent wave is the existence of a proximal to distal gradient in endogenous calcium buffering strength. For example, the delayed appearance of calcium in the distal boutons might reflect a slower rise in free calcium due to greater calcium handling at the boutons. The calcium signals shown in *Figures 5 and 7* suggest that such differences may exist. The signal in the axon initiation zone does not decline during maintained stimulation, whereas the distal signal appears to relax slightly during the stimulation. To test whether slow time-dependent calcium accumulation could account for the delayed rise in distal terminals, paired recordings were made using a 'killswitch' protocol. The stimulation was terminated either prior to (*Figure 8A*) or coincident with (*Figure 8B*) the onset of asynchronous release in ω-conotoxin GVIA-treated fish. In seven such recordings with 5 mM intracellular EGTA, the asynchronous release persisted well beyond termination of the stimulus. To test whether the calcium signal also persists after stimulus termination, we used the killswitch protocol during calcium imaging in the presence of 1 µM ω-conotoxin GVIA. The calcium signal, measured using Fluo-4 in the presence of 0.5 mM EGTA, also exhibited distance-dependent onset with the signal in the distal boutons peaking following termination of the stimulus (*Figure 8C*). Because the stimulus was terminated prematurely in both paired recordings and calcium measurements, continued accumulation would not occur and

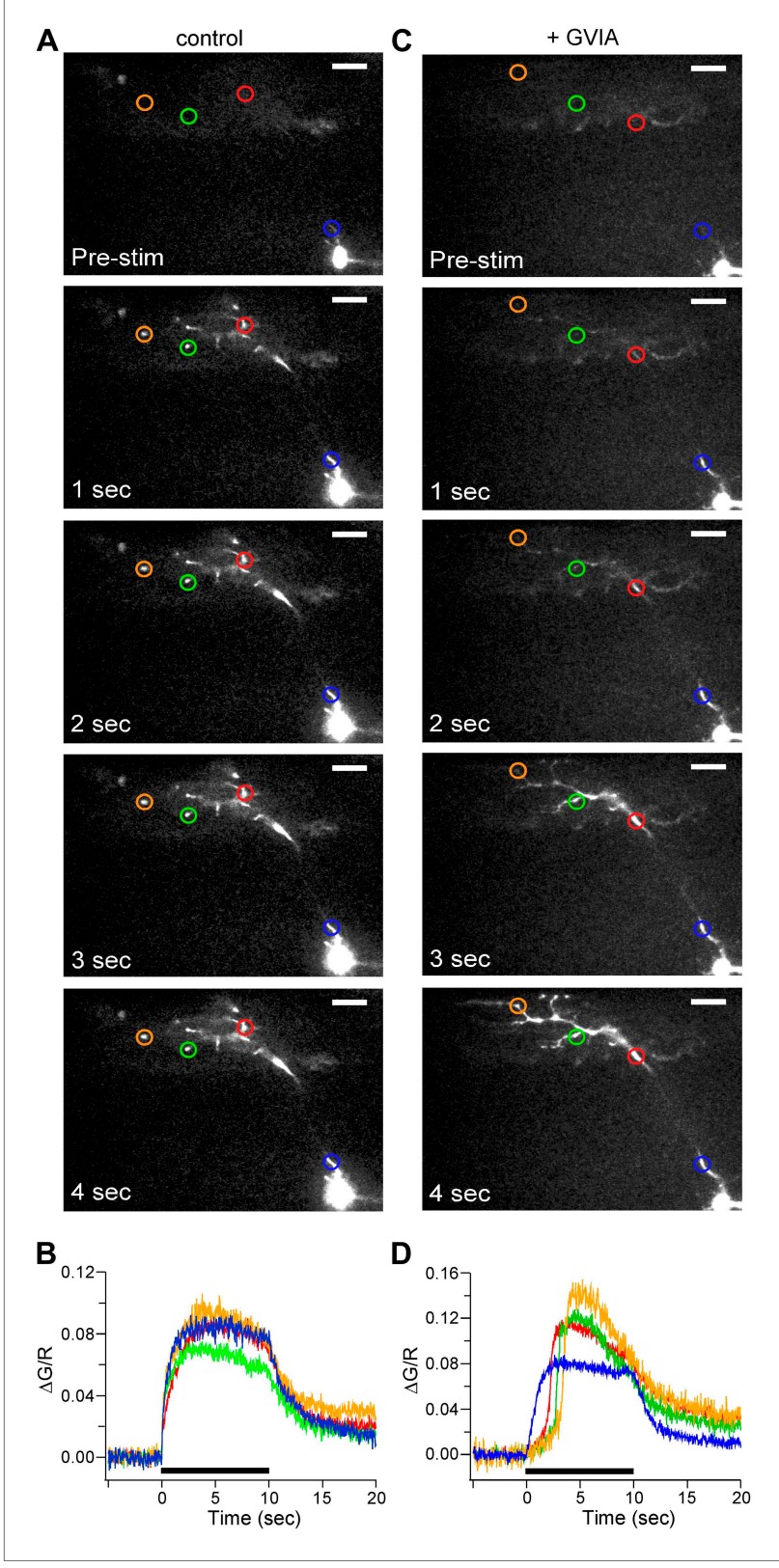

**Figure 5**. Calcium signal onset is delayed at boutons in ω-conotoxin GVIA-treated fish. (**A** and **C**) Sample images of 100 Hz stimulus evoked Fluo-5F fluorescence increases taken at 1 s intervals for 4 ROIs for control (**A**) and ω-conotoxin GVIA-treated (**C**) fish. The scale bar corresponds to 20 µm. (**B** and **D**) The stimulus-driven fluorescence
*Figure 5. Continued on next page*

*Figure 5. Continued*

increases associated with each color-coded ROI in control (**B**) and ω-conotoxin GVIA-treated (**D**) fish. The fluorescence was baseline subtracted and the increase was expressed as ΔG/R. Black bars in (**B**) and (**D**) indicate the timing of 100 Hz stimulation. Experiments were performed with 0.5 mM EGTA in the intracellular solution. The entire videos for **A** and **C** are available as *Video 1* and *Video 2* respectively. For each video the timing of stimulation is indicated by the dot.

could not account for the delayed onset of asynchronous release and the calcium signal. Instead, the findings strongly support the involvement of a propagating calcium signal that was triggered by stimulation at the soma.

As final evidence linking the arrival of calcium to asynchronous release we performed simultaneous paired recording and calcium imaging from ω-conotoxin GVIA-treated fish (*Figure 9*). This further required dye fill of both target muscle and neuron in order to identify the relevant boutons for imaging and measurement of calcium signal. The muscle cell was loaded with Alexa Fluor 555 and the CaP neuron was loaded with Alexa Fluor 647 (*Figure 9A,E*). Stimulus-driven calcium increases (*Figure 9B,F*) were coincident with the postsynaptic asynchronous release measured by means of paired recordings (*Figure 9C,G*). The coincidence between the onset of the calcium signal and asynchronous release was further compared on the basis of synaptic charge entry (*Figure 9D,H*). In three separate simultaneous recordings, the onset time for the calcium signal and asynchronous release, both measured at 50% rise, was within 0.4 ± 0.3 s of each other. Additionally, a large difference in delay of onset was seen between experiments performed with 0.5 mM EGTA (*Figure 9A–D*) and 5 mM EGTA (*Figure 9E–H*) in the internal solution, prompting further investigation into the dependence of the calcium propagation on calcium buffering.

A hallmark of asynchronous release is inhibition by concentrations of the slow calcium buffer EGTA that do not affect synchronous transmission. Using the advantages offered by both calcium imaging (*Figure 10A,B*) and paired recordings of synaptic transmission (*Figure 10C–E*), the effect of intracellular EGTA was determined for both the control and ω-conotoxin GVIA-treated fish. For ω-conotoxin GVIA-treated fish, calcium measurement from ROIs corresponding to boutons showed two to threefold greater delay in signal onset with 5 mM EGTA (red) compared to 0.5 mM EGTA (blue, *Figure 10A*; *Table 1*). The time to reach 20% maximal fluorescence as a function of distance from either the soma or axon reference point was compared for the two EGTA concentrations. Overall, the mean values obtained using 5 mM EGTA were significantly larger at all distances than those obtained using 0.5 mM EGTA for toxin treated fish (*Figure 10B*, p<0.001). Comparing distance-dependent travel of calcium in ω-conotoxin GVIA-treated neurons shows a quasi-linear relationship for the distal boutons (*Figures 7 and 10B*). Fitting distance vs rise in the regions corresponding to boutons to a linear relationship yielded mean slope values corresponding to 74 μm/s and 35 μm/s for 0.5 mM and 5 mM EGTA respectively, consistent with a slower propagating rate in the presence of

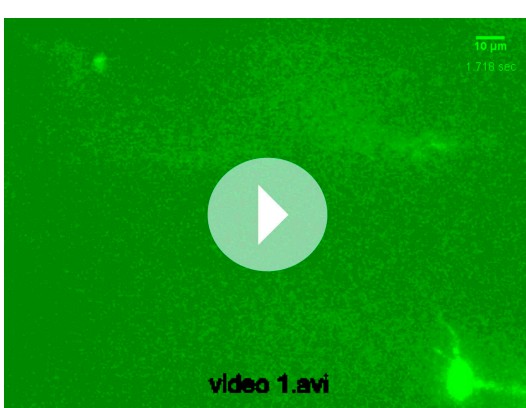

**Video 1**. Control video

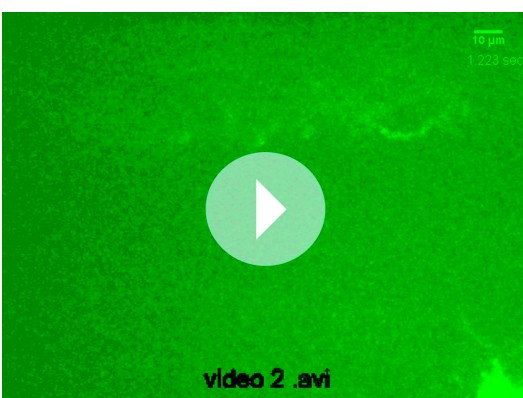

**Video 2**. GVIA video

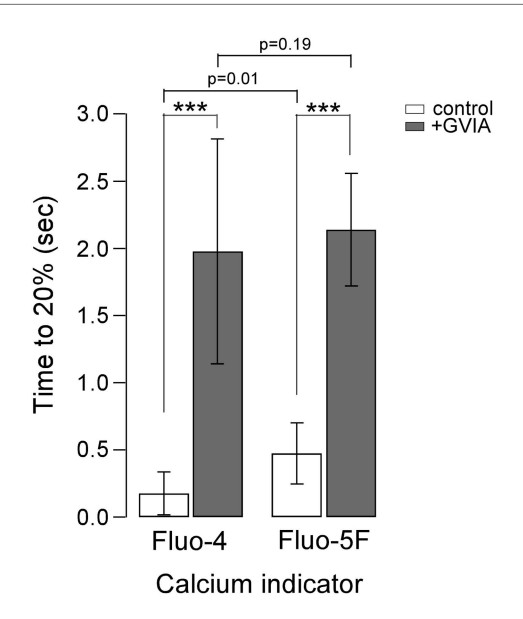

**Figure 6**. The delayed rise of the calcium signal is similar for two different affinity calcium indicators Fluo-4 and Fluo-5F. Comparisons of the time to reach 20% peak stimulated fluorescence in ω-conotoxin GVIA-treated (shaded fill) and control CaP (no fill) motor neurons. In control fish, Fluo-4 onset was 0.18 ± 0.16 s (n = 64 boutons from 4 fish), and Fluo-5F onset was 0.47 ± 0.22 s (n = 91 boutons from 6 fish). For ω-conotoxin GVIA-treated fish, Fluo-4 onset was 1.98 ± 0.84 s (n = 69 boutons from 6 fish), and Fluo-5F onset was 2.14 ± 0.42 (n = 85 boutons from 7 fish). Experiments were performed with 0.5 mM EGTA in the intracellular solution. Asterisks indicate p<0.001.

greater $Ca^{2+}$ buffer (*Figure 10B*). At both the EGTA concentrations, the arrival of the calcium signal in the axonal regions appeared to be independent of distance from the stimulation, despite the large differences in onset time. These data are indicative of a faster velocity of travel along these regions of the neuron. In control fish, the distance-dependence of calcium signal onset was not seen (*Figure 10B*). However, 5 mM EGTA slowed the time to 20% rise in calcium signal compared to 0.5 mM EGTA, reflecting the competition between the indicator and the calcium buffer (*Table 1*, p<0.001).

Paired recordings provided an independent means of comparing the effects of different EGTA concentrations on the onset of asynchronous transmitter release (*Figure 10C–E*). In the presence of ω-conotoxin GVIA, where release is exclusively asynchronous, the time to peak release was greater than twofold larger in the presence of 5 mM EGTA compared to 0.5 mM EGTA (*Table 1*). Sample traces showed the greatly delayed onset of asynchronous release with 5 mM EGTA compared to 0.5 mM EGTA (*Figure 10C*). Inclusion of 25 mM EGTA in the electrode eliminated the majority of synaptic transmission in ω-conotoxin GVIA-treated fish in all but the single recording shown, where the onset of asynchronous release is further delayed compared with those recorded in 5 mM EGTA (*Figure 10C*). Quantitative comparisons of delay were determined on the basis of normalized integrated postsynaptic charge entry vs time from the start of stimulation (*Figure 10D*). The overall data for all of the recordings at 0.5 mM and 5 mM EGTA shows a highly significant difference in the time to peak release (*Figure 10E*). In control recordings the time to peak release was slightly higher for 5 mM EGTA (*Table 1*, p=0.01).

The mechanisms underlying calcium entry and propagation were explored using pharmacological blockers of calcium channels and calcium-operated stores in ω-conotoxin GVIA-treated fish. Treatment of these fish with the non-specific calcium channel blocker $Cd^{2+}$ produced ambiguous results. The high concentrations (500 μM) required to inhibit the propagating calcium signals also inhibited the sodium channels responsible for action potential propagation, so firm conclusions could not be drawn. The rather non-specific T-type calcium channel inhibitors Mibefradil (5 μM) together with $Ni^{2+}$ (100 μM) effectively reduced the calcium signal in the axon and boutons as did the L-type blocker Nimodipine (50 μM). However, other L-type blockers Nitrendipine (25 μM), Nifedipine (50 μM) and Isradipine (50 μM) were without effect. Similarly, the R-type $Ca^{2+}$ channel blocker SNX 482 (200 nM) was completely ineffective. The calcium stores inhibitor thapsigargin (1 μM) and the IP3 receptor antagonists 2-APB (50 μM) and Xestospongin-C (5 μM) were also without effect. Both ruthenium red (400 μM) and ryanodine (100 μM), inhibitors of the ryanodine receptors, gave variable results, only in some cases appearing to increase the arrival time for calcium in the boutons.

Pharmacological treatment with tetrodotoxin (TTX) provided the clearest support for active propagation of the calcium signal. Prolonged depolarization of the CaP motor neuron in the presence of 1 μM TTX resulted in a calcium propagation that was clearly followed from soma to boutons without failure (*Video 3*; *Figure 11A*). In most recordings, multiple propagating signals were observed in the boutons (*Video 3*; *Figure 11B*) and occasionally the signal back-propagated to the soma (*Video 4*) rendering difficult the measurements of velocity. However, in the case shown in *Figure 11* (*Video 3*),

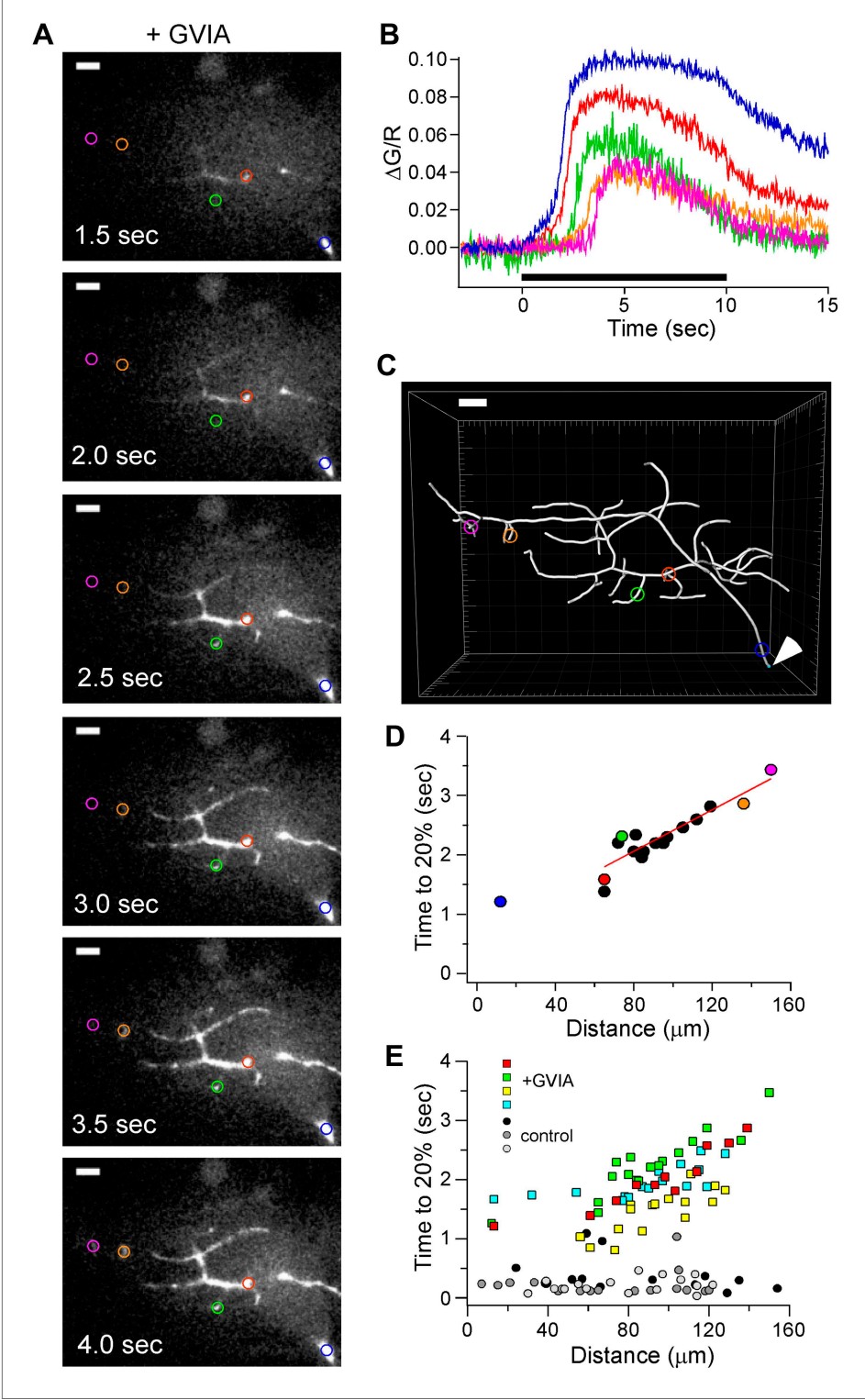

**Figure 7**. Distance-dependent delay in Ca²⁺ rise in ω-conotoxin GVIA-treated CaP boutons. (**A**) Sample images of Fluo-5F fluorescence taken at 0.5 s intervals during 100 Hz stimulation. 5 ROIs are shown at different distances from the reference point at the ventral edge of the notochord. (**B**) The time course of fluorescence change, expressed as ΔG/R, for each of the ROIs shown in **A**. The black bar indicates the duration of stimulation. (**C**) Imaris Filament Tracer 3D reconstruction of the same motor neuron based on z-stacks of the Alexa Fluor 647 fill, with the ROIs in **A** and **B** overlaid. An arrowhead indicates the reference point for distance measurements. Scale bars in **A** and **C** correspond to 10 μm. (**D**)
*Figure 7. Continued on next page*

*Figure 7. Continued*

The time required for each ROI to reach 20% of peak as a function of the distance from the reference point. The distance measurements for each ROI were determined on the basis of Imaris 3D reconstruction. Colored symbols correspond to the individually colored ROIs shown in the **A**–**C**. The data points from the boutons, excluding the first distance measurement, were fit by a line with a slope corresponding to 57 μm/s. (**E**) Scatter plot of distance-dependent Ca²⁺ rise for 61 ROIs in ω-conotoxin GVIA-treated neurons (n = 4 fish, colored markers) and 47 ROIs in control (n = 3 fish, gray markers). Each neuron was reconstructed using Imaris filament software to obtain the physical distances. Example cell in **A**–**D** is shown with green markers. Measurement was obtained with Fluo-5F and 0.5 mM EGTA in the intracellular solution.

the forward propagating velocity for the first propagating signal corresponded to 3 μm/s. Overall estimates fall between 3–10 μm/s that is considerably slower than those obtained with ω-conotoxin GVIA-treated neuron (***Figures 7 and 10***). This difference points to a dominant role of the axonal action potential in coordinating the calcium signaling in the axon initiation zone and the axon proper.

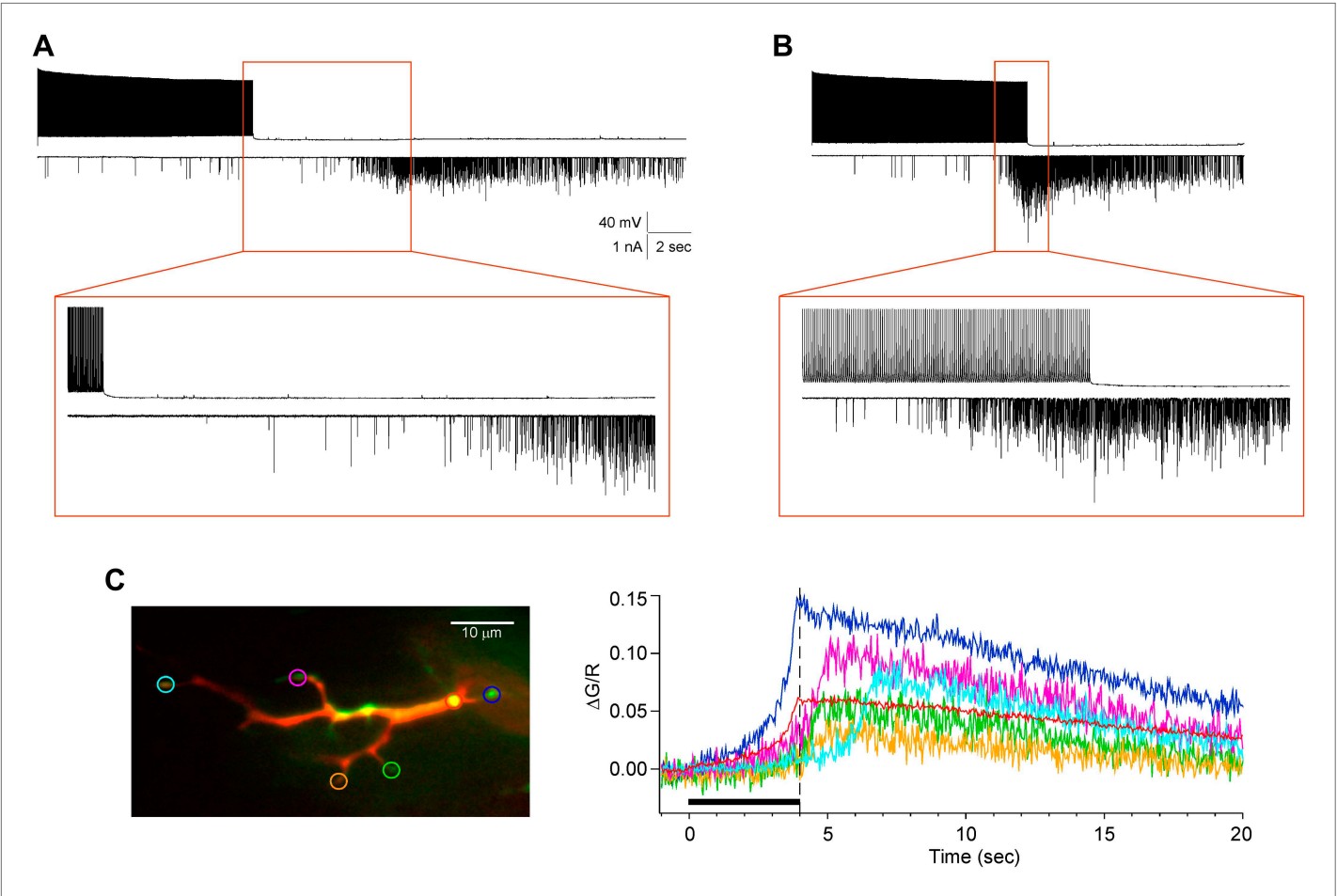

**Figure 8**. The delayed asynchronous release and Ca²⁺ rise in ω-conotoxin GVIA-treated fish is not due to slow local calcium accumulation at the distal boutons. (**A** and **B**) Two paired recordings for killswitch experiments are shown with expanded insets (boxed region). (**A**) In this example, the motor neuron stimulation (top trace) was terminated prior to the sudden onset of asynchronous release (bottom trace). (**B**) In this recording, the stimulus was terminated at the onset of asynchronous transmission, showing the persistence of the release. Both the experiments in (**A**) and (**B**) were performed with 5 mM EGTA in the intracellular solution. (**C**) An example calcium imaging experiment showing that the fluorescence signal peaked after the termination of the 100 Hz stimulation at 4 s. Left: an overlay of single imaging plane with the dye fill (red) and the Fluo-4 signal (green). The individual color coded ROIs were used to generate the associated ΔG/R vs time plot (right panel). The black bar shows the timing of stimulation, with a dashed line indicating the end the stimulation. This experiment was performed with 0.5 mM EGTA in the intracellular solution.

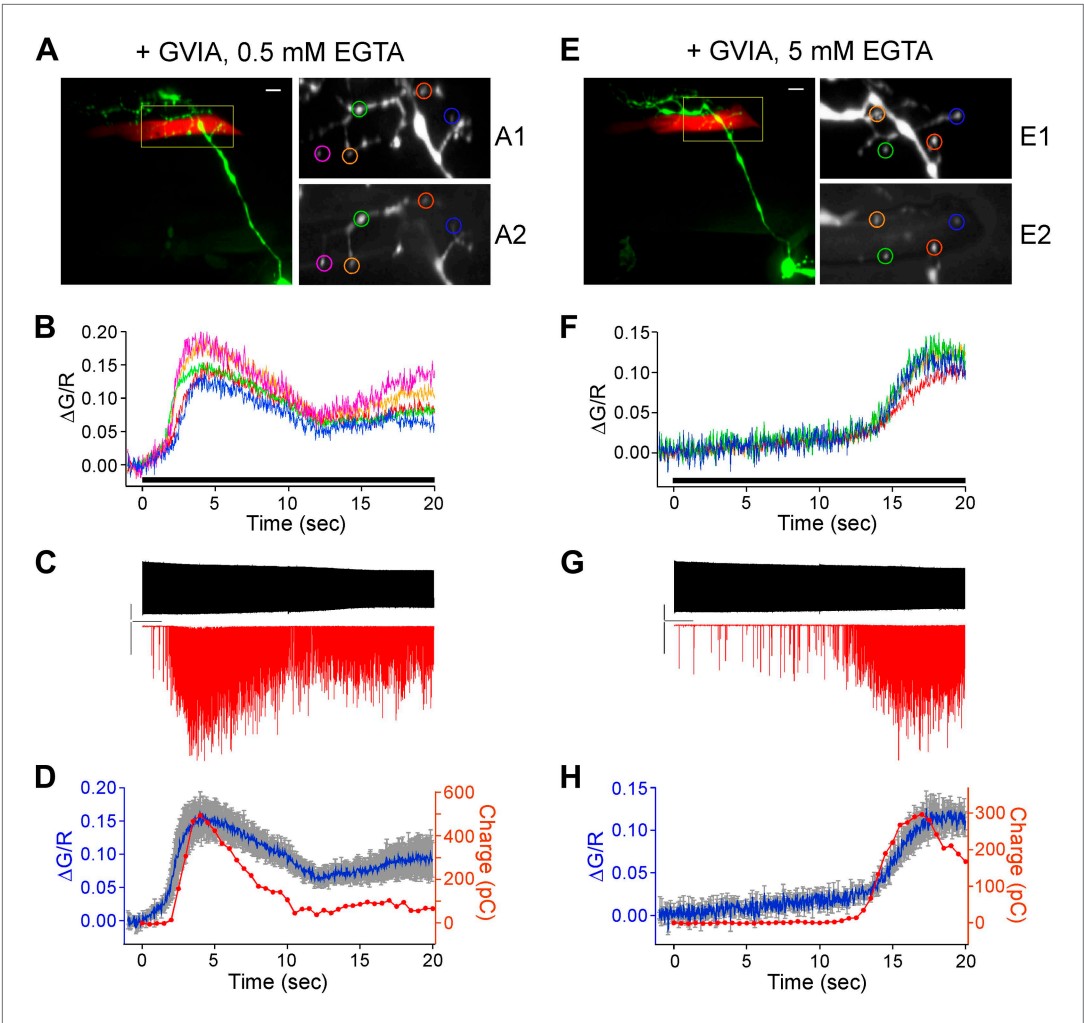

**Figure 9**. Simultaneous paired recordings and calcium imaging in ω-conotoxin GVIA-treated fish. The examples shown compare the effects of 0.5 mM (**A–D**) vs 5 mM (**E–H**) intracellular EGTA. (**A**) and (**E**) Image of the CaP dye filled with Alexa Fluor 647 (green) and target muscle filled with Alexa Fluor 555 (red). Multiple synaptic boutons contacting the target muscle cell are visible as yellow varicosities (scale bar = 10 µm). An enlarged view of the fill (**A1** and **E1**) and peak Fluo-5F calcium response (**A2** and **E2**) are shown for the color coded boutons. (**B**) and (**F**) The associated ΔG/R plots for each of the boutons in **A** and **E** as a function of time during 20 s, 100 Hz stimulus (indicated by black bar), which began at time 0. (**C**) and (**G**) The associated patch clamp recording showing the motor neuron action potential (top) and postsynaptic EPCs (bottom) during the 20 s, 100 Hz stimulation. The scale bars correspond to 40 mV, 1 nA, and 2 s. (**D**) and (**H**) An overlay plot showing the coincidence between the onset of the mean ΔG/R fluorescence for all ROIs (blue with gray SD) and the onset of asynchronous release (red). Release was shown as integrated synaptic charge entry for each consecutive half-second of stimulation.

## Discussion

Paired motor neuron/target fast skeletal muscle recordings have shown that repetitive stimulation of the CaP motor neuron results in a transition from purely synchronous to mixed synchronous/asynchronous release after a delay that is dependent on stimulus frequency (*Wen et al., 2010*). The studies on CNS neurons generally ascribe asynchronous release to persistent calcium resulting from opening of the highly localized presynaptic calcium channels (*Goda and Stevens, 1994*; *Cummings et al., 1996*; *Atluri and Regehr, 1998*; *Chen and Regehr, 1999*; *Lu and Trussell, 2000*). At the zebrafish NMJ, this would be the ω-conotoxin GVIA-sensitive P/Q type calcium channel that is essential for synchronous release (*Wen et al., 2013*). We tested this idea using paired recordings by either eliminating presynaptic

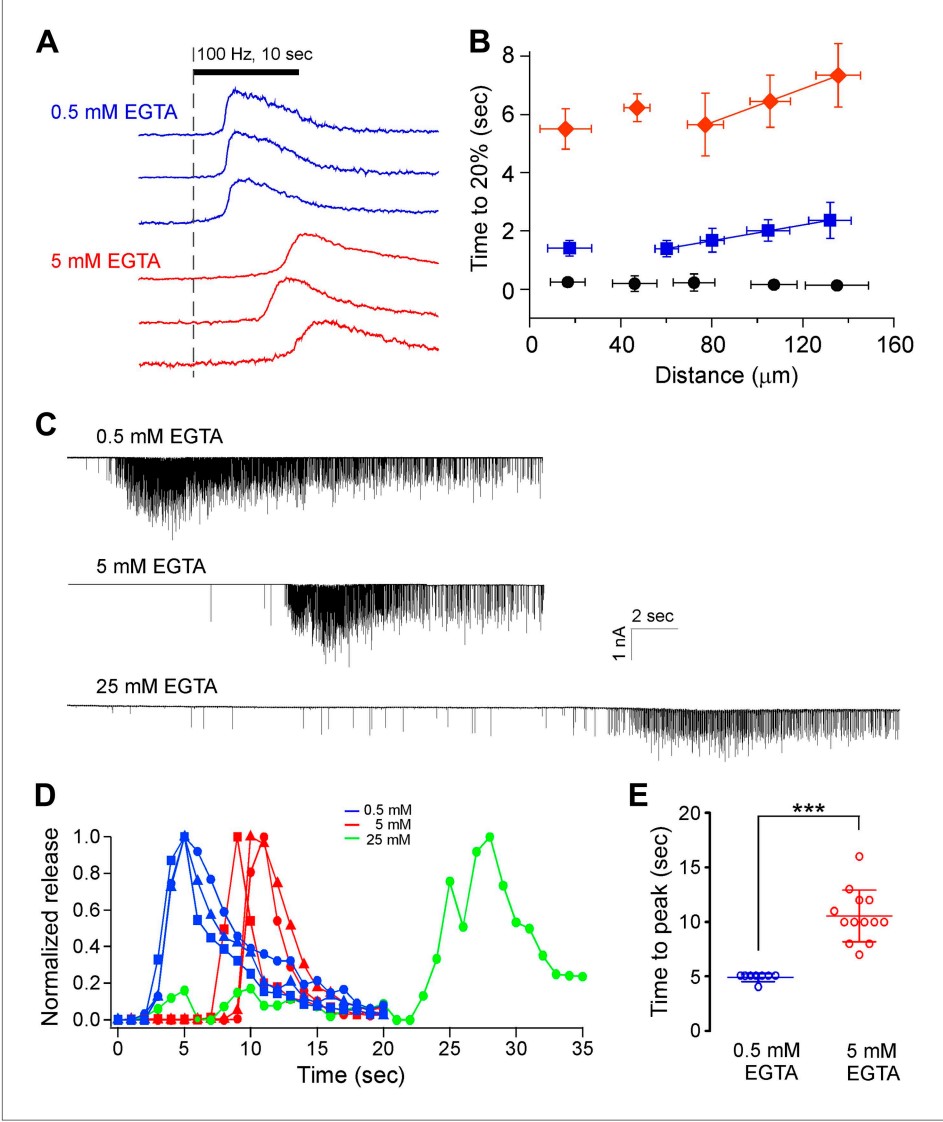

**Figure 10**. Increasing the intracellular concentration of the calcium buffer EGTA further delays both the calcium signal in boutons and the onset of asynchronous release. (**A**) Sample traces comparing the time-dependent Fluo-5F fluorescence increases during stimulation for selected boutons of CaP motor neurons dialyzed with the indicated EGTA concentrations. (**B**) Cumulative data comparing the effects of 0.5 mM (blue; 61 boutons from 4 fish) and 5 mM (red; 44 boutons from 3 fish) EGTA on the time required to reach 20% maximal fluorescence change in boutons of ω-conotoxin GVIA-treated fish. The distances from the reference point were determined using Imaris filament reconstruction and binned (30 μm bin size). Control neuron dialyzed with 0.5 mM EGTA (black; 47 boutons from 3 fish) is shown for comparison. (**C**) Sample patch clamp recordings of muscle EPCs performed at three different EGTA concentrations in ω-conotoxin GVIA-treated fish. The action potentials are not shown but the stimulus lasted 35 s for the bottom trace and 20 s for the other traces. (**D**) The associated integrated synaptic currents as a function of time with 100 Hz stimulation beginning at time 0. Only a single example is shown for 25 mM EGTA because this concentration blocked most transmission in other trials. (**E**) Comparisons of time to reach peak release for all recordings made using 0.5 mM (4.8 ± 0.4 s, n = 8) and 5 mM (10.5 ± 2.4 s, n = 13) EGTA in ω-conotoxin GVIA-treated fish. Data set for 5 mM EGTA was duplicated from *Figure 2* for comparison.

calcium entry through blockade with ω-conotoxin GVIA or by means of a mutant line (*tb204a*) with functionally compromised P/Q type calcium channels (***Wen et al., 2013***). We found that a greatly delayed asynchronous component was still present under conditions where the synchronous release calcium channel was eliminated. Using calcium indicator dyes we identified a source of calcium for

**Table 1.** Comparison of two different intracellular EGTA concentrations on Ca²⁺ signaling and synaptic transmission. The values indicated are all in units of s. Calcium imaging measurements were performed using Fluo-4 indicator.

| | Control | | + ω-conotoxin GVIA | |
| --- | --- | --- | --- | --- |
| | Ca²⁺ imaging (20% rise) | paired recording (time to peak) | Ca²⁺ imaging (20% rise) | paired recording (time to peak) |
| 0.5 mM EGTA | 0.18 ± 0.16 (4 fish, 64 boutons) | 2.88 ± 0.35 (n = 8) | 1.98 ± 0.84 (6 fish, 68 boutons) | 4.88 ± 0.35 (n = 8) |
| 5 mM EGTA | 0.71 ± 0.47 (3 fish, 38 boutons) | 3.36 ± 0.50 (n = 15) | 5.42 ± 1.95 (4 fish, 56 boutons) | 10.54 ± 2.37 (n = 13) |

asynchronous release that, under conditions of P/Q type calcium channel inhibition, originated in the axons and branch points and appeared to propagate into the synaptic boutons. The delayed arrival into the distal boutons, sites of synaptic interaction based on α-btx labeling, accounted well for the delayed onset of asynchronous release. Moreover, paired recordings performed along with calcium imaging confirmed the coincidence between the arrival of the calcium signal and the onset of delayed asynchronous synaptic transmission.

The delay in the onset of asynchronous release was longest in ω-conotoxin GVIA-treated fish, intermediate in mutant *tb204a* fish and shortest in wild-type fish. This rank order was inversely related to the levels of P/Q type channel function, pointing to a likely contribution by these channels to asynchronous release as well. Accordingly, the early onset of asynchronous release in control recordings, compared to ω-conotoxin GVIA-treated fish, would result from the combined action of the local calcium entry and calcium arrival from a distal source. The synergistic action is fully compromised in ω-conotoxin GVIA-treated fish but only partially compromised in mutant fish. A model with separate, but interactive calcium sources, challenges the prevailing view that asynchronous release results exclusively from an expansion of the highly restricted calcium domains surrounding the same calcium channels that govern synchronous release (**Borst and Sakmann, 1996**; **Meinrenken et al., 2002**). This model would predict that blocking the source of calcium for synchronous release would also inhibit the asynchronous release. Support for an expanding domain comes principally from the differential sensitivity of synchronous vs asynchronous release to slow calcium buffers (**Neher, 1998**; **Eggermann et al., 2012**). The slow calcium buffer EGTA obliterates asynchronous release at concentrations that are ineffective on synchronous release (**Adler et al., 1991**; **Cummings et al., 1996**; **Atluri and Regehr, 1998**; **Lu and Trussell, 2000**). By contrast, blockade of synchronous release generally requires use of the fast buffer BAPTA (**Adler et al., 1991**). Augustine notes that 'this criteria has been used successfully to probe local calcium signaling with the conclusion that high BAPTA/EGTA efficacy points toward nanodomain and equal efficacy implicates diffuse calcium signals or microdomains' (**Augustine et al., 2003**), an approach widely used as a determinant of calcium channel-calcium sensor coupling (**Eggermann et al., 2011**). It is the case with zebrafish NMJ that asynchronous release, as well as the calcium propagation, is highly sensitive to the slow buffer EGTA. We interpret these findings to mean that in zebrafish the differential sensitivity reflects, to a large extent, the involvement of bulk cytoplasmic changes stemming from distal calcium as opposed solely to highly restricted calcium domains entering through the P/Q type calcium channels. This provides an alternative interpretation to the BAPTA/EGTA sensitivity metric but does not exclude additional contribution by the expanding zone of calcium formed by local P/Q channel openings.

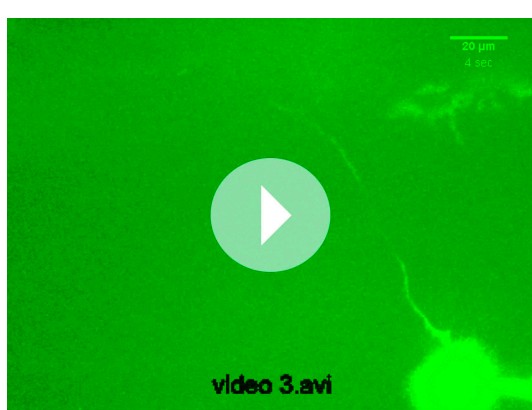

**Video 3.** TTX1

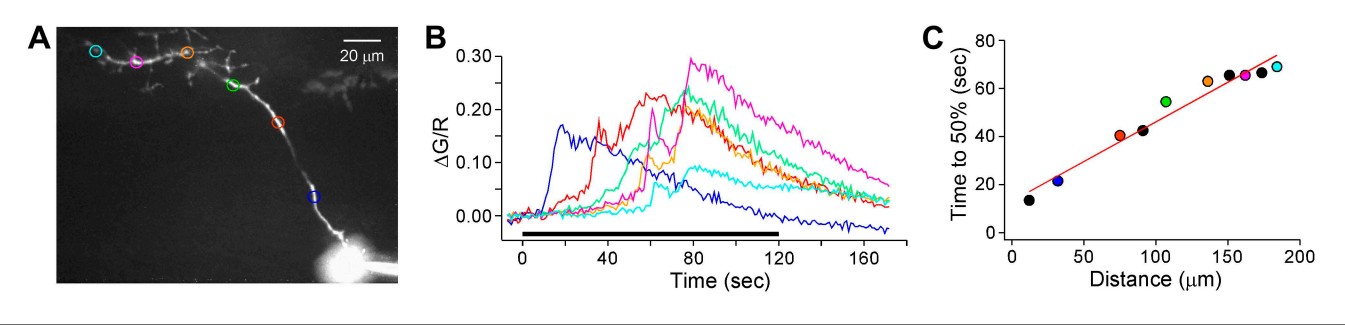

**Figure 11**. Depolarization-induced regenerative calcium wave in the presence of TTX. (**A**) Peak response of the Fluo-4 calcium signal following depolarization from −80 mV to +50 mV for 2 min in the presence of 1 μM TTX. The color coded ROIs are indicated. (**B**) Measurements of ΔG/R from the ROIs indicated in (**A**) show that calcium propagation traveled in two successive, distinct movements into the boutons. Each data point represents the flattened stack of 15 sections, each 1 μm thick, acquired with 1 s intervals between stacks. Experiment was performed with 0.5 mM EGTA in the intracellular solution. (**C**) Time to reach 50% of the first peak was plotted against distance from the soma for the example. ROIs shown in **A** and **B** are indicated with colors and additional ROIs not shown are black. The data points were fit to a line. The video for this recording and a second example recording are available as *Video 3* and *Video 4* respectively.

The source of the axonal calcium entry underlying the propagation and the mechanism by which calcium propagates in a regenerative fashion are both unknown. The pharmacological results point to no clear source of either intracellular or extracellular calcium as required for the initiation or propagation. Effects seen using the calcium channel blockers, Mibefradil, Ni²⁺ and Nimodipine suggest that activation of calcium channels is involved, but the pharmacological profile does not fit any specific isoform. One impression resulting from these experiments is that the propagation is so potent in the branch points and locations near boutons, that there may be a large safety factor that requires near extinction of calcium before propagation is terminated. In this case, the pharmacological block would have to be nearly complete before the calcium propagation was inhibited. Our results from TTX experiments provide some insights into the process of calcium propagation. In the absence of action potentials, $Ca^{2+}$ propagation can be elicited with prolonged depolarization, but the apparent velocity is slow. These data are compatible with a model in which $Ca^{2+}$ propagation is less dependent on coordination by the action potential in the boutons than in the axons. Accordingly, an action potential traveling down the axon would instantaneously initiate $Ca^{2+}$ propagation at the branching point, whereas the spread of depolarization would take much longer in the presence of TTX. The pharmacological conditions under which the calcium propagation is best revealed are non-physiological. Instead, the distance over which calcium needs to propagate under physiological conditions is certainly less due to participation of the action potential. Additionally, as pointed out earlier the P/Q channels do participate in the calcium generation and the secondary sources for calcium are also near the boutons, minimizing the need for long distance propagation. However, all evidence points to a central involvement of this secondary source of calcium in the delayed onset of asynchronous release.

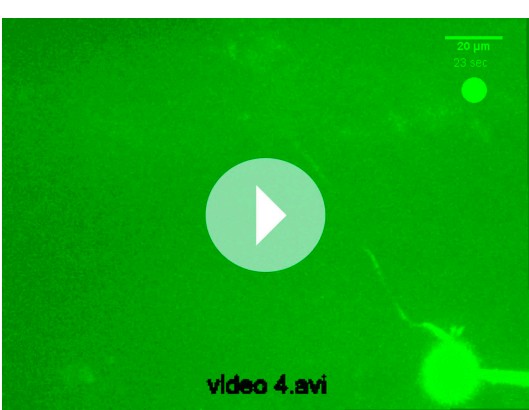

**Video 4**. TTX2

Our finding of a robust propagating calcium signal in the motor neuron was quite unexpected. Published accounts of propagating calcium signals like those shown in our study have been largely limited to postsynaptic cells (***Ross, 2012***). However, there is some precedence for ryanodine-sensitive calcium stores involvement in synaptic transmission at guinea pig sympathetic nerve terminals and frog NMJ (***Smith and Cunnane, 1996***; ***Narita et al., 1998***). Our findings take on additional significance due to our ability to assign a

functional role to this secondary source of calcium. Consequently, we have a clearer understanding of the delayed onset and persistent release properties associated with asynchronous release. In light of the ubiquitous nature of asynchronous release, it will be critical to determine whether similar mechanisms are at work at synapses within the central nervous system.

## Materials and methods

Zebrafish (*Danio rerio*) were maintained in accordance with the standards set forth in the International Animal Care and Use Committee. Brian's wild-type strain and the *tb204a* mutant line (**Wen et al., 2013**) were used for all experiments. The electrophysiology and imaging experiments were performed exclusively on larva between the ages of 72–96 hr post-fertilization (hpf). Methodology for mounting and preparing the fish for paired recordings are detailed in the video publication (**Wen and Brehm, 2010**). The intracellular solution contains (in mM): 115 K-gluconate, 15 KCl, 2 $MgCl_2$, 10 K-HEPES, 4 Mg-ATP, pH 7.2 with 0.5 mM EGTA or 5 mM EGTA as indicated with each experiment. The extracellular solution contains (in mM): 134 NaCl, 2.9 KCl, 1.2 $MgCl_2$, 2.1 $CaCl_2$, 10 glucose, 10 Na-HEPES, pH 7.8.

For calcium imaging, either 100 μM Fluo-4 or 100 μM Fluo-5F was loaded into the CaP neuron by means of the recording patch pipette. Live confocal images were acquired using a Yokogawa CSU-10 spinning disc (Yokogawa, Tokyo, Japan) with a Stanford Photonics 620 Turbo ICCD camera (Stanford Photonics, Palo Alto, CA). The laser lines used included an Argon 488 nm (DLS2000, Dynamic Laser, Salt Lake City, UT), 561 nm (Sapphire 561-50 CW CDRH, Coherent, Santa Clara, CA) and 640 nm (Chromalase II diode laser, Blue Sky Research, Milpitas, CA). Low power images that included the entire CaP motor neuron utilized the Zeiss Plan-Apochromat 40×/1.0 n.a. dip objective, whereas higher resolution of the bouton field was obtained with the 63x equivalent objective. The CaP motor neuron was also co-loaded with 40 μM Alexa Fluor 647 hydrazide (Invitrogen, Eugene, OR) to identify the boutons and perform morphological reconstructions. For paired recordings with simultaneous calcium imaging, muscle was filled with 40 μM Alexa Fluor 555 hydrazide (Molecular Probes, Eugene, OR). For each synaptopHluorin or calcium indicator recording, an acquisition plane was selected to contain 10–15 synaptic boutons in the field. Sequential images were acquired continuously at 33 ms intervals during 100 Hz stimulation. The calcium signals for each bouton were baseline subtracted, and the ratio of stimulus induced increase in calcium signal (green) to fill signal (red) was computed for regions of interest (ROIs). Images were acquired with Piper Control 2.5.04 (Stanford Photonics, Palo Alto, CA) and analyzed with ImageJ (NIH, Bethesda, MD), Microsoft Excel (Redmond, WA) and custom scripts in Igor Pro 6.3 (Lake Oswego, OR). Unlike the calcium images taken in a single plane, images of the Alexa Fluor 647 filled motor neuron were acquired before and after the end of the experiment using 50–70 1 μm steps (Mipos 100SG piezo driver; Piezosystems Jena, Jena, Germany) to create stacks for the purpose of full morphological determination. The dye filled CaP motor neuron was reconstructed using Imaris filament software (Bitplane, Zurich, Switzerland) after completion of the calcium imaging.

The *Tol2* transposon system was used to generate the transgenic fish line expressing synaptopHluorin in the CaP motor neuron. SynaptopHluorin was constructed by fusing super-ecliptic pHluorin (**Miesenbock et al., 1998**) in frame to C-terminus of the zebrafish Vamp2 with an 8-amino acid linker. It was cloned into pTol2000 vector with the Huc promoter for primary motor neuron expression. The plasmid DNA was co-injected with in vitro synthesized transposase mRNA into one cell embryos. Founder fish were screened by PCR and fluorescence microscopy for expression in the spinal motor neurons.

All data are presented as mean ± SD and statistical comparisons were made using standard *t* tests. TTX, ω-conotoxin GVIA and α-btx were obtained from Alomone Labs (Jerusalem, Israel). Fluo-4, Fluo-5F, Alexa Fluor dyes were obtained from Invitrogen (Eugene, OR). Ruthenium red, ryanodine, Mibefradil, Nimodipine, and $Ni^{2+}$ were obtained from Sigma-Aldrich (St. Louis, MO). Nitrendipine, Nifedipine, Isradipine, SNX 482, thapsigargin, 2-APB, and Xestospongin-C were obtained from Tocris Bioscience (Bristol, UK).

## Acknowledgements

The authors thank Aurelie Synder and the Jungers Advanced Imaging Center for training with Imaris. Dr George Peeters of Solamere Technology Group (Salt Lake City, UT) helped design the imaging hardware and analysis software.

## Additional information

### Funding

| Funder | Grant reference number | Author |
|---|---|---|
| National Institutes of Health | NS082573 | Hua Wen, Jeffrey M Hubbard, Benjamin Rakela, Paul Brehm |
| Muscular Dystrophy Association | MDA236717 | Hua Wen, Paul Brehm |
| Howard Hughes Medical Institute | | Gail Mandel |
| Muscular Dystrophy Association | MDA255543 | Michael W Linhoff |

The funders had no role in study design, data collection and interpretation, or the decision to submit the work for publication.

### Author contributions

HW, Conception and design, Acquisition of data, Analysis and interpretation of data, Drafting or revising the article; JMH, Conception and design, Acquisition of data, Analysis and interpretation of data; BR, Acquisition of data, Analysis and interpretation of data; MWL, GM, Conception and design, Drafting or revising the article; PB, Conception and design, Analysis and interpretation of data, Drafting or revising the article

### Ethics

Animal experimentation: All of the experiments in this study utilized zebrafish larvae (*Danio rerio*) under the age of 7 days. The protocols and care were carried out in accordance with the Guide for Care and Use of Laboratory Animals of the National Institutes of Health. Care for the animals was under the governance of the Department of Comparative Medicine at OHSU. Additionally, all protocols were reviewed and approved by the institutional IACUC committee (IS0002667).

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
