## [Decision Letter]

Thank you for sending your work entitled “Synchronous and asynchronous modes of synaptic transmission utilize different calcium sources” for consideration at *eLife*. Your article has been favorably evaluated by a Senior editor and 2 reviewers, one of whom is a member of our Board of Reviewing Editors.

The Reviewing editor and the other reviewer discussed their comments before we reached this decision, and the Reviewing editor has assembled the following comments to help you prepare a revised submission.

This provocative manuscript describes an unusual phenomenon that the authors have discovered in zebrafish motor neurons. When P/Q-type Ca channels are blocked with conotoxin, presynaptic activity still results in neurotransmitter release, but after an extremely long delay. A nice set of electrophysiology and imaging experiments make it pretty convincing that this unexpected phenomenon is real, and involves strangely delayed calcium rises in response to activity. It raises the possibility that this phenomenon could exist at other synapses, which makes it interesting. On the down side, the properties of this phenomenon are only sketched in here, with a lot of loose ends, and the exposition is somewhat disordered. Furthermore, the terminology presumes identity of this phenomenon with other established phenomena such as delayed release and calcium waves, even though there is little evidence to support this. These weaknesses detract from what is otherwise an interesting observation, quite worthy of more thorough investigation. Minimally, substantial editorial changes in the text, figures, and data analysis are required. It is possible that some additional experiments will be needed, or addition of further data already in place in the laboratory may suffice.

Major comments:

1) There are some unnecessary distractions in terminology that go beyond the data as presented here. First, calling this “delayed” or “asynchronous” release is confusing. Delayed release at conventional central synapses is the desynchronized release of individual quanta following calcium influx at synapses, and is driven to a large extent by residual calcium. This phenomenon studied here may be different in important ways. It is difficult to tell from the figures whether these are individual quanta, since they seem to be very large currents. Perhaps enlarging and analyzing the traces in Figure 1 would help. What is the quantal size at this synapse? Are these multiple quanta releasing simultaneously, or are quanta actually 1-2 nA? Secondly, the authors argue that this calcium influx is not through normal synaptic calcium channels, so tying this phenomenon to delayed release seems unnecessary. Second, the release follows an elevation in intracellular calcium that is referred to as a “wave”. This term brings to mind the spontaneous, repetitive activity seen in developing cochlea and retina. However, the phenomenon studied here is not clearly spontaneous or repetitive, so it would be preferable to avoid using the loaded term “wave”. I think this phenomenon is quite interesting enough on its own terms to avoid the confusion associated with those established phenomena, until it can be clearly linked to them. It seems adequate to call it “propagating”, as for example an action potential is.

2) In contrast to what the authors write, the data appear to show a clear disparity between asynchronous release and the slow calcium wave in the EGTA experiments. While increasing [EGTA] leads to a consistent increase in the onset of the [Ca^2+^] wave, the onset of the asynchronous release first decreases significantly before increasing back to the original size:

No EGTA: time to peak of normalized release = 10.5 s time to 20% of [Ca^2+^] peak = 2.1 s

0.5 mM EGTA: time to peak of normalized release = 4.8s time to 20% of [Ca^2+^] peak = ∼2 s

5.0 mM EGTA: time to peak of normalized release = 10.5 s time to 20% of [Ca^2+^] peak = ∼6 s

These numbers are hidden in the manuscript, as the authors do not include the relevant control data (P/Q blocked, but no EGTA) in the graphs. It is left to the reader to pick out these data from other figures.

Moreover, it is not clear how to reconcile the different time courses of calcium rises vs release across the dataset. Calcium appears to rise rapidly (2-3 s, Figures 4 and 6) in some experiments, and slowly in others (Figures 7 and 9). Likewise, release rises rapidly in some experiments (Figure 7), but not others (Figures 1, 2, 8 and 9). What is going on here? Figure 7 links rise of Ca and release to EGTA concentration. Does this account for the variability in different figures? If so, please give the EGTA concentration in every figure. If it doesn't, then what is going on? Figure 9 gives us a single example of similarity between rate of rise of Ca and release. Were there any other examples? I appreciate that this experiment may have been difficult, but a single N seems too little. In addition, Figure 9 leaves it to the reader to draw the correlation by eye. Can the authors take a more quantitative approach here?

3) Related to the above point, there appears to be some data duplication (presumably unintentional): Figure 1, the data points labelled “+GVIA” are identical with the data points shown in Figure 7 as “5 mM EGTA”. This needs to be corrected.

4) There are a number of features to this phenomenon that are simply not clearly studied here. Figures 7 and 8 raise the possibility that much of the stimulation has no effect on rises in Ca or release. This is confusing. The calcium signal in Figure 7 clearly decreases while stimulation is ongoing. Is the signal spontaneously ending, or is it depleted? How long does recovery take? There is one tantalizing remark that the signal bounced “occasionally”. This sounds very anecdotal. Could this be replicated? Why is it not shown? Furthermore, the data of Figure 7 do not seem appropriate for linear fits, because there is a large jump in latency close to the soma. This does not seem consistent with the idea of a “wave” of calcium emanating from the soma. In any case, those linear fits, if made, should be shown on the figure. Why is the role of different calcium sources not studied? What happens if other Ca channel blockers are applied (N-type, R-type)? What happens if store blockers are applied?

5) The “killswitch” experiment (Figure 8) is elegant, but it doesn't show what the authors conclude from it (namely that it proves the involvement of a propagating calcium wave). Rather it shows that onset of asynchronous release doesn't require ongoing stimulation. While this result is suggestive, the authors need to actually show that the calcium wave also persists after termination of the stimulation in this protocol to draw their conclusion. This is a doable experiment.

---

## [Author Response]

*1) There are some unnecessary distractions in terminology that go beyond the data as presented here. First, calling this “delayed” or “asynchronous” release is confusing. Delayed release at conventional central synapses is the desynchronized release of individual quanta following calcium influx at synapses, and is driven to a large extent by residual calcium*.

We considered the asynchronous “jitter” reported by others to potentially be accounted for by offsite calcium, but we have absolutely no evidence that this is the case. Consequently, we removed all reference to this form of delay and only use the term in the context of the greatly delayed onset of asynchronous release following the onset of stimulation under conditions wherein synchronous release is blocked.

*This phenomenon studied here may be different in important ways. It is difficult to tell from the figures whether these are individual quanta, since they seem to be very large currents. Perhaps enlarging and analyzing the traces in*
Figure 1
*would help. What is the quantal size at this synapse? Are these multiple quanta releasing simultaneously, or are quanta actually 1-2 nA*?

Indeed the large currents are individual quanta, as previously reported for this synapse. To further aid in addressing this important issue we added amplitude histograms for evoked asynchronous and spontaneous synaptic currents to Figure 1 as suggested.

*Secondly, the authors argue that this calcium influx is not through normal synaptic calcium channels, so tying this phenomenon to delayed release seems unnecessary. Second, the release follows an elevation in intracellular calcium that is referred to as a “wave”. This term brings to mind the spontaneous, repetitive activity seen in developing cochlea and retina. However, the phenomenon studied here is not clearly spontaneous or repetitive, so it would be preferable to avoid using the loaded term “wave”. I think this phenomenon is quite interesting enough on its own terms to avoid the confusion associated with those established phenomena, until it can be clearly linked to them. It seems adequate to call it “propagating”, as for example an action potential is*.

It could be argued that the term “wave” would be appropriate in light of the ability of the calcium signal to propagate both anterograde and retrograde without decrement as reflected in the new TTX figure. However, to err on the side of conservative, we refer to the phenomenon as “calcium propagation” as suggested.

*2) In contrast to what the authors write, the data appear to show a clear disparity between asynchronous release and the slow calcium wave in the EGTA experiments: While increasing [EGTA] leads to a consistent increase in the onset of the [Ca*^*2+*^*] wave, the onset of the asynchronous release first decreases significantly before increasing back to the original size*:

*No EGTA: time to peak of normalized release = 10.5 s time to 20% of [Ca*^*2+*^*] peak = 2.1 *s

*0.5 mM EGTA: time to peak of normalized release = 4.8s time to 20% of [Ca*^*2+*^*] peak = ∼2 *s

*5.0 mM EGTA: time to peak of normalized release = 10.5 s time to 20% of [Ca*^*2+*^*] peak = ∼6 *s

*These numbers are hidden in the manuscript, as the authors do not include the relevant control data (P/Q blocked, but no EGTA) in the graphs. It is left to the reader to pick out these data from other figures*.

*Moreover, it is not clear how to reconcile the different time courses of calcium rises vs release across the dataset. Calcium appears to rise rapidly (2-3 s,*
Figures 4 and 6*) in some experiments, and slowly in others (*Figures 7 and 9*). Likewise, release rises rapidly in some experiments (*Figure 7*), but not others (*Figures 1, 2, 8 and 9*). What is going on here?*
Figure 7
*links rise of Ca and release to EGTA concentration. Does this account for the variability in different figures? If so, please give the EGTA concentration in every figure. If it doesn't, then what is going on?*
Figure 9
*gives us a single example of similarity between rate of rise of Ca and release. Were there any other examples? I appreciate that this experiment may have been difficult, but a single N seems too little. In addition,*
Figure 9
*leaves it to the reader to draw the correlation by eye. Can the authors take a more quantitative approach here*?

This was all very confusing and lacked proper annotation so we apologize. We have addressed this point of confusion by now including the EGTA concentration in all figures as requested. We also performed additional EGTA experiments so that we could compile side-by-side comparisons for 5 mM and 0.5 mM EGTA using both calcium imaging and paired recordings. These data are now presented in a new Table 1. The reviewers correctly note the differences in calcium rise time but this represents the systematic differences between low and high EGTA concentration.

*3) Related to the above point, there appears to be some data duplication (presumably unintentional):*
Figure 1*, the data points labelled “+GVIA” are identical with the data points shown in*
Figure 7
*as “5 mM EGTA”. This needs to be corrected*.

The duplication was intentional for the basis of comparison and is now noted in the figure legend.

*4) There are a number of features to this phenomenon that are simply not clearly studied here.*
Figures 7 and 8
*raise the possibility that much of the stimulation has no effect on rises in Ca or release. This is confusing. The calcium signal in*
Figure 7
*clearly decreases while stimulation is ongoing. Is the signal spontaneously ending, or is it depleted? How long does recovery take? There is one tantalizing remark that the signal bounced “occasionally”. This sounds very anecdotal. Could this be replicated? Why is it not shown? Furthermore, the data of*
Figure 7
*do not seem appropriate for linear fits, because there is a large jump in latency close to the soma. This does not seem consistent with the idea of a “wave” of calcium emanating from the soma. In any case, those linear fits, if made, should be shown on the figure. Why is the role of different calcium sources not studied? What happens if other Ca channel blockers are applied (N-type, R-type)? What happens if store blockers are applied*?

- First, the calcium signal does plateau and often declines during prolonged stimulation. As the decline occurs for boutons and not the axon, this could signal differences in calcium handling for boutons and axons. We note this in the results along with our rationale for the killswitch experiment.

- Second, we did not look at recovery due to the need for holding pairs for such a long period of time.

- Third, the active propagation, multiple wave phenomenon and bounce back in TTX is now fully documented in Figure 11. Two movies are shown along with an example figure. The example shown in the figure exhibited two waves. This clear example was chosen because it helped resolve the issue of propagation velocity. The second example, shown in movie form, demonstrates forward and back propagation of the calcium signal (Video 4: TTX2). In the presence of TTX the velocity of propagation is slowed and the fast propagation seen in the axon of GVIA treated fish is missing. We discuss the importance of this distinction in the Results.

- Fourth, we added the fit over the linear range to Figures 7 and 10. We also explain our reasoning as to why we excluded the axon points from the conotoxin treated neuron. In fitting the TTX data we included all of the points and we also provided an interpretation for the differences in these data sets in the Results.

- Fifth, we added a paragraph in the Results summarizing the pharmacological reagents tested and the associated findings. We also included a brief presentation about the sources of calcium in the Discussion.

*5) The “killswitch” experiment (*Figure 8*) is elegant, but it doesn't show what the authors conclude from it (namely that it proves the involvement of a propagating calcium wave). Rather it shows that onset of asynchronous release doesn't require ongoing stimulation. While this result is suggestive, the authors need to actually show that the calcium wave also persists after termination of the stimulation in this protocol to draw their conclusion. This is a doable experiment*.

The doable experiment was indeed doable, but it wasn’t easy. The results have been added to Figure 8.